# Yellow Sticky Cards Reduce the Numbers of *Trichogramma dendrolimi* (Hymenoptera: Trichogrammatidae) Following Augmentative Releases against the Fruit Borers *Carposina sasakii* (Lepidoptera: Carposinidae) and *Grapholita molesta* (Lepidoptera: Tortricidae) in a Pear Orchard

**DOI:** 10.3390/insects15080590

**Published:** 2024-08-03

**Authors:** Lu Gan, Yanan Wu, J. P. Michaud, Yisong Li, Xiaoxia Liu, Songdou Zhang, Zhen Li

**Affiliations:** 1Department of Entomology and MOA Key Lab of Pest Monitoring and Green Management, College of Plant Protection, China Agricultural University, Beijing 100193, China; ganlu014456@163.com (L.G.); yanan_wo@126.com (Y.W.); liuxiaoxia611@cau.edu.cn (X.L.); sdzhang2013@cau.edu.cn (S.Z.); 2Department of Entomology, Kansas State University, Agricultural Research Center-Hays, Hays, KS 67601, USA; jpmi@ksu.edu; 3Key Laboratory of the Pest Monitoring and Safety Control of Crops and Forests, College of Agriculture, Xinjiang Agricultural University, Urumqi 830052, China; lysexb@sina.com

**Keywords:** biological control, egg parasitoid, chromatotropism, Chrysopidae, Coccinellidae

## Abstract

**Simple Summary:**

Laboratory and field observations were used to evaluate the compatibility of yellow sticky cards with augmentative releases of *T. dendrolimi* for control of orchard pests, particularly *C. sasakii* and *G. molesta*. Female *T. dendrolimi* showed stronger attraction to yellow and white than to red, green or blue. Yellow sticky cards trapped significant numbers of *T. dendrolimi* in the field, reducing biological control of *C. sasakii* as well as the number of lacewings naturally recruited to the orchard. Judicious use of yellow sticky cards in orchards is therefore recommended to avoid disruption of biological control, specifically removing them in May before peak flights of moth pests require releases of egg parasitoids.

**Abstract:**

Integrated pest management relies upon mutual compatibility among pest control tactics. The fruit-boring moths *Carposina sasakii* and *Grapholita molesta* can be devastating pests of pome and stone fruit production. *Trichogramma dendrolimi* parasitizes the eggs of these pests, preventing their eclosion, but its efficacy can be reduced by other pest control tactics. We tested *T. dendrolimi* attraction to five colors, and moth attraction to six colors, in laboratory choice tests, and thereafter deployed yellow sticky cards in tandem with releases of *T. dendrolimi* in field trials in a pear orchard. Yellow sticky cards deployed at high density trapped *T. dendrolimi* and reduced their numbers post-release. They also trapped adult *G. molesta*, which appeared to compensate for reduced egg parasitism on this species, but not on *C. sasakii*, which had higher abundance in plots with yellow sticky cards. The cards also captured adult lacewings, likely reducing their numbers in the field, but did not capture large numbers of lady beetles. The results suggest that yellow sticky cards can be used at high density to control aphids, psyllids and leafhoppers in early spring (March and April) when natural enemies are in low numbers, then removed in May so as not to interfere with augmentative releases of *T. dendrolimi* that must be timed to coincide with peak flights of fruit-boring moths. This strategy should enhance the compatibility of yellow sticky cards with egg parasitoid releases.

## 1. Introduction

Integrated pest management (IPM) is widely regarded as an effective approach to achieve sustainable pest control [1,2,3]. In contrast to dependence on chemical pesticides, IPM promotes the use of combinations of compatible pest control tactics, which include plant resistance and physical, biological and chemical interventions [4,5]. However, some combinations of pest control tactics prove incompatible, and can even exacerbate pest problems via antagonistic effects. Therefore, a prerequisite for effective IPM is a careful assessment of the compatibility of pest control tactics to be used simultaneously [6,7,8]. Biological control, whether achieved via conservation measures or augmentation of specific agents, is a foundational component of agricultural IPM. However, many parasitoids, predatory mites and entomopathogenic fungi are tiny organisms that are extremely sensitive to environmental conditions [9,10,11], and incompatible with other pest control tactics such as chemical pesticides and insect pathogens [12,13,14]. For example, Garzón A. et al. (2015) found sulfoxaflor to be slightly toxic to adult green lacewings, *Chrysoperla carnea* (Stepphens) (Neuroptera: Chrysopidae), but very toxic to fourth instar larvae of the two-spot lady beetle *Adalia bipunctata* L. (Coleoptera: Coccinellidae). However, deltamethrin was highly toxic to larvae and adults of both species [15]. Furthermore, high temperatures may synergize pesticide toxicity for predators [16,17], by virtue of combining two stress factors [1]. Microbial agents such as entomopathogenic fungi not only infect and kill targeted pests, but also have the potential for pathogenicity to non-target arthropods, especially those with broad host ranges [18,19,20,21]. For example, application of *Metarhizium* strains at 10^8^ conidia mL^−1^ killed 27.5–67.5% of *Menochilus sexmaculatus* (Fabricius) (Coleoptera: Coccinellidae) larvae and reduced adult emergence by 3–30% [22]. Some physical control tactics may also diminish natural enemy effectiveness, such as nets installed to exclude pests from protected cultures but that also prevent the immigration of natural enemies [23,24].

Colored sticky cards are another tactic widely used for physical control and monitoring that takes advantage of the specific chromatotropisms expressed by insect pests. Yellow sticky cards are commonly used and a key component of IPM programs for various small pests like aphids, psyllids and midges in greenhouses and fruit orchards, given their simplicity and low cost for long-term and large-scale applications [25,26,27]. However, field studies have demonstrated that sticky cards also trap and kill various beneficial insects along with pests, sometimes at even higher rates [27,28,29]. For instance, Zhang G. et al. (2020) found that the number of natural enemies trapped by yellow sticky cards in apple orchards was almost equal to the number of target pests, *Aphis citricola* Van der Goot (Homoptera: Aphididae), and that the cards were also detrimental to bees [27]. Similarly, Chen H. et al. (2012) found that yellow sticky cards not only trapped aphid pests in orchards, but also hymenopterous parasitoids [29]. The ecological impacts of yellow sticky cards remain ambiguous in many pest control contexts, and their compatibility with natural enemy augmentation programs requires clarification in specific situations.

In orchards, environmentally friendly tactics are increasingly preferred, and compatibility among pest control methods is a prerequisite for improving fruit yield and quality. The peach fruit moth, *Carposina sasakii* Matsmura (Lepidoptera: Carposinidae) and the oriental fruit moth, *Grapholita molesta* (Busck) (Lepidoptera: Tortricidae), are both devastating pests of pome and stone fruits in the family Rosaceae, such as apple, pear, peach and plum. Larvae of both species bore into fruits soon after eclosion and feed internally until pupation, causing serious economic losses [30,31,32,33]. Larvae of *G. molesta* also bore into new shoots, impeding fruit tree growth and development. The concealed feeding locations of larvae make these pests difficult to control, and their regional impact is expected to worsen as global warming extends growing seasons and enables additional generations [30,31,34]. At present, pheromone-based mating disruption and chemical applications are widely used for control of both pests in fruit orchards [35,36,37,38,39]. However, continuous exposure to high concentrations of sex pheromones has led to reduced efficacy of mating disruption tactics [35,36,40], and long-term reliance on pesticides for control has resulted in the pests evolving resistance to various modes of action [37,38,39].

Control of the egg stage is critical for management of borer pests, as the eggs are usually exposed on the surface of plants. *Trichogramma* spp. parasitoids have been successfully released against boring pests of rice, corn, and sugarcane [41,42,43]. Augmentative releases of *Trichogramma dendrolimi* Matsumura (Hymenoptera: Trichogrammatidae) have shown notable efficacy against *G. molesta* in peach and pear orchards [44,45], but their compatibility with other orchard pest control tactics has not been rigorously assessed. One tactic of potential concern is the use of yellow sticky cards to trap aphids and other small orchard pests, as these can also attract adult Hymenoptera [46,47,48].

The present study had three objectives: (1) to clarify the attractiveness of yellow sticky cards (and other colors) to *T. dendrolimi*, *C. sasakii* and *G. molesta* in behavioral assays in the laboratory; (2) to assess the potential impact of yellow sticky cards on the biocontrol efficacy of *T. dendrolimi* augmented against fruit borers in a pear orchard by continuously monitoring insect populations; (3) to explore how the combined use of yellow sticky cards and *T. dendrolimi* augmentation affects the population dynamics of natural enemy insects in the orchard. The results were expected to clarify whether yellow sticky cards were compatible with augmentation of *T. dendrolimi* for control of *C. sasakii* and *G. molesta* in an orchard setting and assist in the development of a rational IPM plan for control of orchard pests.

## 2. Materials and Methods

### 2.1. Insect Rearing

A laboratory colony of *G. molesta* was established from about 300 larvae collected from a pear orchard at the Liaoning Fruit Research Institute (120°7′ E, 40°6′ N) in 2018 and reared in the Integrated Pest Management (IPM) laboratory of China Agricultural University. The colony was continuously reared for more than 30 generations in a climate-controlled chamber (Ningbo Saifu, Ningbo, China) under standardized conditions of 25 ± 1 °C, 70 ± 5% RH, and a 14:10 (L:D) photoperiod. Developing larvae were reared on Fuji apples and transferred to Petri dishes (9.0 cm diam) covered with four layers of gauze after they emerged from fruit as mature fifth instars. Pupae were then sexed and transferred to ventilated disposable plastic boxes (14 cm × 10 cm × 8 cm) purchased online, 30 individuals per box (20 males and 10 females), and provisioned with a 10% honey solution on balls of cotton. After adult emergence and mating, eggs were laid on the inner surfaces of the boxes. Moths were transferred to new boxes daily to enable collection of eggs; boxes bearing eggs were cut into pieces and placed next to fresh apples. Eclosing larvae would then immediately bore into fruit to feed and complete development.

A laboratory colony of *C. sasakii* was established from about 500 eggs provided by the Research Institute of Pomology of CAAS (120°4′ E, 40°4′ N) in 2018 and reared in the Integrated Pest Management Laboratory of China Agricultural University. The colony was reared continuously for more than 20 generations in a climate-controlled chamber (Ningbo Saifu, China) under the standardized conditions (as for *G. molesta*). Larvae were reared on Fuji apples and, after they emerged from fruit as mature fifth instar, were transferred to ventilated plastic boxes (14 cm × 10 cm × 8 cm) containing slightly moistened sawdust as a pupation substrate. Pupae were sexed and transferred to clean boxes (as above), 21 individuals per box (14 males and 7 females), and provisioned with a 10% honey solution on balls of cotton [49,50]. Procedures for egg collection were identical to those used for *G. molesta*.

Eggs of the rice moth, *Corcyra cephalonica* (Stainton) (Lepidoptera: Pyralidae), were used as a host for rearing *T. dendrolimi*. About 500 eggs of *C. cephalonica* were obtained from the Biological Control Institute of Jilin Agricultural University, and reared on an artificial diet prepared with corn flour, wheat bran, yeast, and sugar, as previously described [51], for more than 30 generations in a climate-controlled chamber under the same conditions as *G. molesta*. Briefly, eggs of *C. cephalonica* were sprinkled on the surface of the artificial diet in ventilated plastic boxes (40 cm × 25 cm × 15 cm), and covered with a thin layer of the diet, after which the box was covered with 2 layers of gauze. After eclosion, larvae fed on the diet until pupation and emergence as adults. The adults were then transferred to cages (27 cm × 24 cm × 15 cm), and eggs were collected and transferred to fresh containers with diet [51]. To prepare egg cards, one-day-old eggs of *C. cephalonica* were stuck to paper cards (4 cm × 1 cm, ca. 800 eggs per card) using non-toxic, double-sided tape. To arrest embryonic development, egg cards were irradiated at a distance of 15 cm from a 30 W ultraviolet lamp (Beijing Guangda Hengyi Technology Co., Ltd., Beijing, China) for 1 h on an ultra-clean laboratory bench (Beijing Yatai Cologne Instrument Technology Co., Ltd., Beijing, China). The egg cards so prepared were used as hosts for rearing *T. dendrolimi*.

*Trichogramma dendrolimi* were purchased from Jilin Gongzhuling Jinong Green Agriculture High-tech Co., Ltd. (Changchun, China), where they are mass-produced in eggs of *Antheraea pernyi* (Guerin-Meneville) (Lepidoptera: Saturniidae). Our laboratory colony of *T. dendrolimi* was propagated on *C. cephalonica* eggs for more than 30 generations before use in experiments, and mated females (<24 h old) were used in all laboratory trials.

### 2.2. Choice Tests of Light Intensity and Color with T. dendrolimi

Because light intensity affects *T. dendrolimi* behavior, we first tested the responses of *T. dendrolimi* females to five light intensities. Thereafter, female responses to five different colors were compared under the most preferred light intensity. All tests were performed in three replicates in circular plastic arenas (28 cm diam × 12.0 cm ht) covered with transparent film and divided radially into five equal parts using opaque cards, with a circular central area in which wasps were released. 

The light intensity test was conducted in a growth chamber under 1400 lux light and set to the standard rearing conditions described above. The side walls of each arena (n = 3) were covered with opaque black paper and tops of the five sections were covered with 0, 1, 3, 5 and 7 layers of gauze, respectively, to create light intensities of 1400 lux, 1000 lux, 600 lux, 500 lux and 400 lux (measured with a light intensity meter, Hansha Scientific Instruments Co., Ltd., Taian, China). The bottom and side surface of the choice sections of the arena were covered with double-sided tape to capture wasps which entered each of the five sections. A *T. dendrolimi* card, containing ca. 800 wasp pupae, was placed in the central circle of each arena for wasp emergence. About 5 days later, after all wasps had emerged and died, dishes were opened to observe and record the number of *T. dendrolimi* captured in the five sections under different light intensities. 

For the color preference assay, the side walls of the five sections of each arena (n = 3) were covered with different colors of paper purchased online. The colors were determined according to the international hexadecimal color code (white #FFFAFA, red #EA2A37, yellow #E3DE00, green #71B03D and blue #215BFB). The top of each section was covered with a transparent film, and the arenas were placed in a growth chamber under the same conditions as described above, under a light intensity of about 1000 lux. Releases of *T. dendrolimi* were conducted as described for the light intensity assay.

### 2.3. Color Choice Tests with C. sasakii and G. molesta

Choice tests were conducted in nylon cages (1 m × 1 m × 1 m, n = 3 replicates per moth species), with sticky cards (21 cm × 29.7 cm) of six different colors (white, red, yellow, green, blue and black) hanging from the roof in random arrangement (Appendix A). In each test, 25 2-day-old adult *C. sasakii* or *G. molesta* were released in the center of the cage on the bottom; a 10% honey solution was provided on cotton balls. The cages were placed in a growth chamber under the standard condition (as above), until all moths had either died or landed on a sticky card. The cage was then opened, and the number of moths trapped on each sticky card was tallied. Subsequently, based upon the results from the 6-way color choice test, we then conducted two-way choice tests for each species using the two most preferred colors, with two sticky cards of different colors placed at opposite sides of the cage wall (Appendix A).

### 2.4. Monitoring the Orchard Insect Community

Field observations were conducted in a pear orchard located in Dingfuzhuang Village, Panggezhuang Town, Daxing District, Beijing (116°45′ E, 39°98′ N) in 2021. The pear orchard was about 54 hectares in area, and consisted primarily of *Pyrus pyrifolia* cv. Cuiyu. The trees were 8 years old with an average height of 2.5 m, spaced 0.7 m apart in rows 3.0 m apart. During the period of observation, the main insect pests were pear psylla, *Cacopsylla chinensis* (Yang et Li) (Hemiptera: Psyllidae), and oriental fruit moth, *G. molesta*. The pesticides spiroteramat, emamectin benzoate, abamectin, pyridaben and thiamethoxam were applied as required at various phenological stages of the fruit crop, as described in previous work [45]. 

The population dynamics of *T. dendrolimi* and its hosts *C. sasakii* and *G. molesta*, along with other natural enemy species, were continuously monitored in the orchard from late April to mid October in 2021. Yellow sticky cards were used to monitor *T. dendrolimi* and other natural enemies, and traps with pheromone lures were used to monitor the target pests *C. sasakii* and *G. molesta*. Yellow sticky cards and pheromone lures and traps were purchased from Beijing Zhongjie Sifang Biotechnology Co., Ltd. (Beijing, China). Egg cards of *T. dendrolimi* for release in the orchard were purchased from Beijing Yihuan Natural Enemy Agricultural Technology Co., Ltd. (Beijing, China), which rears them on eggs of *A. pernyi*.

The orchard was divided into three rectangular sections, each separated by a buffer of 15 m. In each section, three blocks of pear trees (1350 m^2^ each) were established as experimental replicates (Figure 1). The three northern blocks were assigned as controls (CK), and the other six blocks were alternately assigned to one of two treatments, either *T. dendrolimi* releases alone (Td), or *T. dendrolimi* releases plus adjacent yellow sticky cards (Y + Td). Each block contained six yellow sticky cards (21 × 29.7 cm) plus three pheromone traps for monitoring the insects. Each trap contained one lure for *C. sasakii* and one lure for *G. molesta*. 

Both the yellow sticky cards and the pheromone traps were attached to pear trees at a height of 1.5 m and spaced as depicted in Figure 1. The sticky cards and trap sheets in the pheromone traps were both replaced weekly, and all insect data recorded; pheromone lures were replaced monthly. At peaks of *G. molesta* emergence, determined by pheromone trap catches and data from previous years, 60,000 *T. dendrolimi* (as mature pupae in *A. pernyi* eggs) were released at the center of each block on 22 June, 6 July, 27 July and 17 August, respectively, in 2021.

### 2.5. Data Analysis

All the data were recorded using Microsoft Excel v. 16.7 (Microsoft, 2010, Redmond, WA, USA), the statistical analysis was performed with SPSS 20.0 (IBM Corporation, 2011, Chicago, IL, USA), and graphs were drawn using GraphPad Prism 6 (GraphPad Software, 2018, San Diego, CA, USA). The laboratory choice tests among different light intensities and colors by *T. dendrolimi*, and choice tests among sticky cards of different colors by *C. sasakii* and *G. molesta*, were analyzed by Chi-square Goodness-of-fit test, with the null hypothesis of an equal distribution across different light intensities and colors. Choice tests with yellow and white sticky cards for *C. sasakii* and *G. molesta* were analyzed by paired *t*-test (two-tailed). The field trap catches of insects in different treatments within the pear orchard were compared by one-way ANOVA followed by Tukey’s HSD mean separation test after square-root transformation of the data with the formula of $\sqrt{x+0.5}$ (where *x* represents the catch number) [52], and a test for homoscedasticity of variance.

## 3. Results

### 3.1. Light Intensity and Color Choice Tests with T. dendrolimi

Adult *T. dendrolimi* preferred regions of the arena with higher light intensities (*χ*^2^ = 58.269, *df* = 4, *p* < 0.001). The largest percentage chose the section under 1000 lux, significantly more than chose 400 or 500 lux, but not significantly more than chose 600 or 1400 lux (Figure 2A). Therefore, all further choice tests were conducted under 1000 lux.

The color choice test revealed that *T. dendrolimi* preferred white and yellow over red, green, or blue (*χ*^2^ = 39.091, *df* = 4, *p* < 0.001; Figure 2B). 

### 3.2. Color Choice Tests with G. molesta and C. sasakii

Both *C. sasakii* and *G. molesta* preferred white and yellow cards over cards of other colors (*χ*^2^ = 87.356, *df* = 5, *p* < 0.001; Figure 3A, B). In subsequent two-way choice tests, *C. sasakii* preferred white to yellow sticky cards (*t* = 13.86, *df* = 2, *p* = 0.005), whereas *G. molesta* showed no preference (*t* = 2.60, *df* = 2, *p* = 0.122; Figure 3C).

### 3.3. Field Observations

Natural populations of *T. dendrolimi* were at low levels in mid-June, but increased significantly after each augmentative release (Figure 4A). The numbers of *T. dendrolimi* caught on monitoring sticky cards were significantly lower in plots where yellow sticky cards were deployed as traps than in those they were not (*F*_2,6_ = 2179.213, *p* < 0.001; Figure 4D).

Two generations of *C. sasakii* were observed in the pear orchard. Overwintered adults began to appear in mid-May in 2021, and the population remained low until emergence of the first generation in early August, with peak flight occurring around mid- to late August (Figure 4B). Releases of *T. dendrolimi* did not significantly reduce the numbers of *C. sasakii* caught relative to control plots, but more were caught in blocks with yellow sticky cards than in those where *T. dendrolimi* was augmented without them (*F*_2,6_ = 6.628, *p* = 0.030; Figure 4E). Therefore, the yellow sticky cards reduced the effective population of augmented *T. dendrolimi*.

Four generations of *G. molesta* were detected in the orchard, with the overwintering adults emerging in late March and peaking in late April. The first generation peaked in early June at a lower population density than the overwintering cohort, and later generations appeared from late July to late September (Figure 4C). Augmentation of *T. dendrolimi* significantly reduced the *G. molesta* population (*F*_2,6_ = 14.619, *p* = 0.005), and plots with high densities of yellow sticky cards did not have greater numbers of *G. molesta* than plots without them (Figure 4F).

### 3.4. Recruitment of Natural Enemies

The three main groups of natural enemies observed in the orchard were lady beetles (Coleoptera: Coccinellidae), hoverflies (Diptera: Syrphidae) and lacewings (Neuroptera: Chrysopidae). Lady beetles began to appear on sticky cards in late April, but had disappeared by early September, with the population peaking around mid- to late June (Figure 5A). There was no significant difference among treatments in the numbers of lady beetles trapped (*F*_2,6_ = 1.177, *p* = 0.371; Figure 5D).

Hoverflies were observed from mid-April to early October, with population peaks in mid-June and late August (Figure 5B). There was no significant difference among treatments in the numbers of flies trapped (*F*_2,6_ = 0.224, *p* = 0.806; Figure 5E).

Lacewings were observed from mid-April to early October, although relatively few in number, and without any obvious peak in abundance (Figure 5C). Releases of *T. dendrolimi* had no significant effect on lacewing catches, but a high density of yellow sticky cards significantly reduced trap catches (*F*_2,6_ = 21.513, *p* = 0.002; Figure 5F). 

## 4. Discussion

Insects have evolved chromatotropism in various forms to aid in habitat location and foraging [53,54]. Colored sticky cards have been developed that take advantage of these responses to trap pests [55]. However, colored sticky cards can have inadvertent side effects, such as trapping parasitic and predatory insects, potentially compromising their compatibility with IPM programs. The present study revealed that *T. dendrolimi* oriented preferentially to yellow and white, and that high densities of yellow sticky cards deployed to trap orchard pests will inadvertently trap *T. dendrolimi* and are likely to reduce the efficacy of wasps released in augmentation programs against boring lepidopterous pests. In addition, yellow sticky cards also reduced densities of lacewings naturally recruited to the orchard. 

The observed preference of *T. dendrolimi* for yellow and white over other colors is shared by many hymenopteran egg parasitoids, and may reflect chromatotropism associated with either parasitism behavior (pale or white host eggs) or feeding behavior (yellow flowers as nectar sources). For example, female *Trichogramma ostriniae* Pant et Chen prefer yellow and white model eggs, and exhibit differences in searching behavior and parasitism among model eggs of different colors [56]. *Trichogrammatoidea bactrae* Nagaraja prefers white flowers to those of other colors, as they tend to serve as a reliable source of nectar [57]. Therefore, the potential impacts of colored sticky cards on other naturally occurring parasitoids should be carefully considered, especially in orchard IPM programs that rely on conservation biological control or use intercropped flowers to provide supplementary nutrition for natural enemies [11,58]. 

The pear orchard in the study had a history of significant fruit boring damage by *C. sasakii* and *G. molesta*, and augmentative releases of *T. dendrolimi* have provided measurable benefits [45]. However, pear orchards also suffer damage from aphids, leafhoppers and psyllids in spring, and yellow sticky cards are deployed at high densities to trap and kill these pests. In our laboratory choice tests, *C. sasakii* and *G. molesta* also oriented preferentially to yellow and white, indicating another potential benefit of yellow sticky cards. Other work [59] has demonstrated that a yellow substrate will shorten preoviposition periods and enhance fecundity in both *C. sasakii* and *G. molesta*, findings consistent with our results. These collective benefits of yellow traps also must be balanced against their potential to reduce the biocontrol efficacy of *T. dendrolimi* releases and the cost of alternative controls. 

Field observations conducted in the growing season of 2021 revealed that yellow sticky cards caught large numbers of *T. dendrolimi*, reducing their numbers significantly compared to plots where wasps were released without the cards. Also, more *C. sasakii* were caught in the former plots, suggesting the suppressive effect of *T. dendrolimi* releases on this species was also reduced. In contrast, catches of *G. molesta* appeared unaffected by yellow sticky cards, suggesting that *T. dendrolimi* releases were equally effective against this species in their presence. However, *G. molesta* appeared more strongly attracted to yellow than *C. sasakii* in laboratory choice tests, so an increased efficacy of yellow sticky traps against this species may have mitigated a lower rate of egg parasitism. Nevertheless, catches of *C. sasakii* in yellow sticky cards did not appear to compensate for the reduced control resulting from catches of *T. dendrolimi*. Therefore, the compatibility of *T. dendrolimi* releases with simultaneously deployed yellow sticky cards will depend to some extent on the chromatotropic responses of the target pests. 

Naturally occurring beneficial insects trapped on yellow sticky cards included lady beetles, hoverflies and lacewings, the lady beetles being the most abundant group. Although numbers of lady beetles and hoverflies were not significantly reduced by yellow sticky cards relative to plots without them, numbers of lacewings were. Previous studies indicate that sticky cards do not catch many lacewings, and that their responses to color are inconsistent. Whereas Wu et al. found that lacewings were more attracted to yellow sticky cards than those of other colors [60], other studies have reported only slightly higher catches on yellow versus other colors, or no significant difference [61,62]. Clearly, more studies are warranted to assess the potential impacts of yellow sticky cards on natural enemies in orchard settings, as the existing literature reports effects that range from insignificant to negative [27,63,64]. 

## 5. Conclusions

Both laboratory and field data indicated that yellow sticky cards are attractive to *T. dendrolimi* and could reduce their numbers following augmentative releases in a pear orchard. In the case of *G. molesta*, the negative effect of the sticky cards on parasitoid numbers may have been mitigated by the trapping of adult moths. Nevertheless, appropriate scheduling of these two pest control tactics in an orchard pest management program could improve their compatibility. Yellow sticky cards can be deployed in high density in early spring (March to April) to control aphids, psyllids and leafhoppers [65,66], as natural enemy populations are low at this time. Beginning in May, yellow sticky cards can be removed so that they do not interfere with augmentative releases of *T. dendrolimi* that are timed to coincide with peak flights of fruit-boring moths. This strategy should enhance overall management of the pest complex that consists of sucking hemipteran pests early in the growing season, and fruit-boring moth pests that occur later. 

## Figures and Tables

**Figure 1 insects-15-00590-f001:**
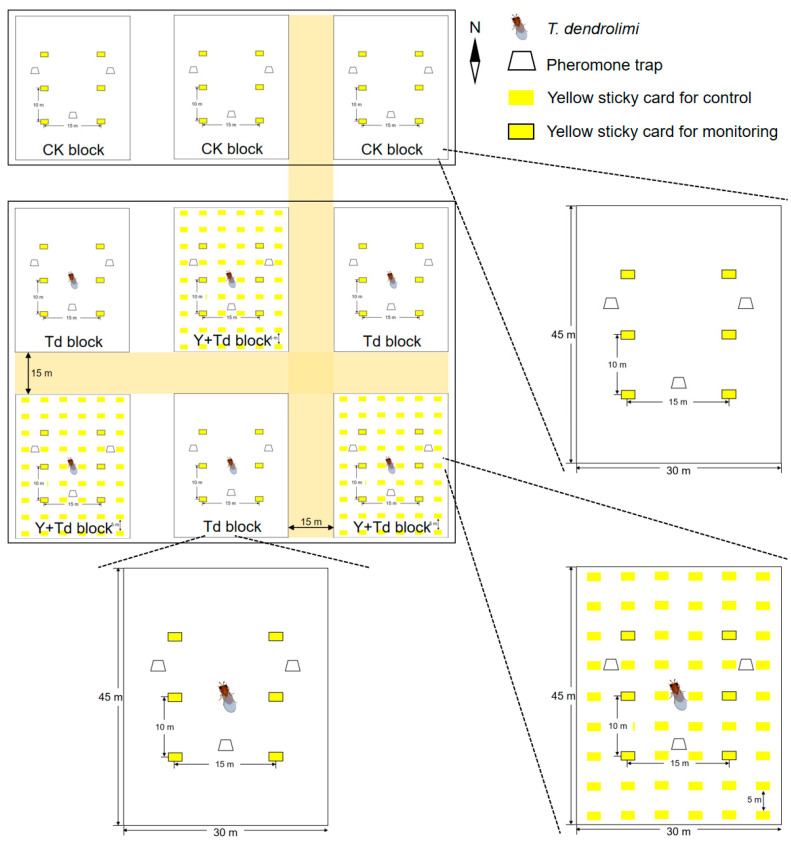
Schematic diagram of the field experiment layout in the pear orchard depicting the arrangement of treatment blocks with *Trichogramma dendrolimi* release cards, yellow sticky cards, and pheromone traps. The three northern blocks were assigned as controls (CK), and the other six blocks assigned to receive either *T. dendrolimi* releases alone (Td), or *T. dendrolimi* releases plus adjacent yellow sticky cards (Y + Td).

**Figure 2 insects-15-00590-f002:**
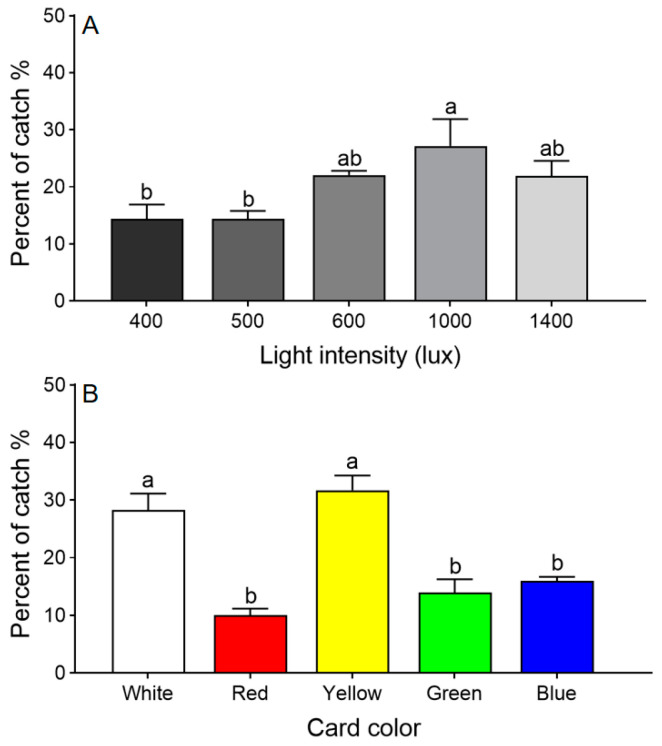
Choice tests with *T. dendrolimi* adults responding to different light intensities (**A**) and sticky card colors (**B**). Means with different letters above error bars indicate significant differences among different treatments (Chi-square Goodness-of-fit test, α = 0.05).

**Figure 3 insects-15-00590-f003:**
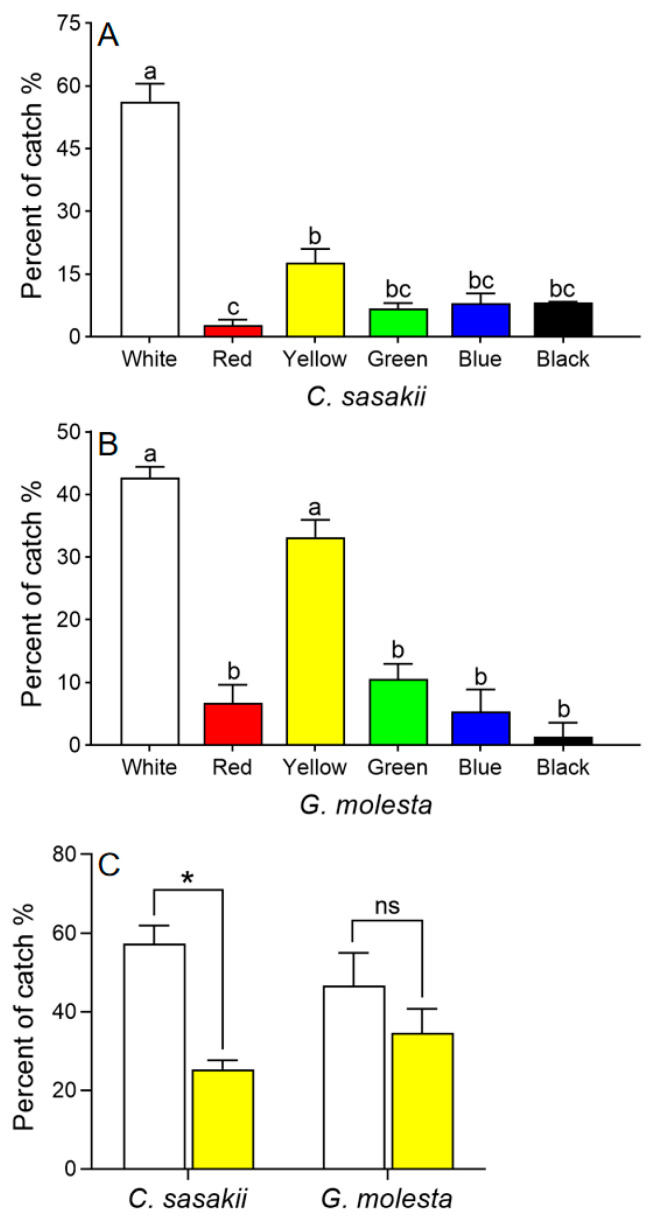
Choice tests with *Carposina sasakii* (**A**) and *Grapholita molesta* (**B**) adults responding to different colored sticky cards. Means with different letters were significantly different among cards of different colors (Chi-square Goodness-of-fit test, α = 0.05). (**C**) Choice test with *C. sasakii* and *G. molesta* choosing between yellow and white sticky cards (paired *t*-test (two-tailed), α = 0.05; * represents significant difference between the two colors, ns indicates no significant difference).

**Figure 4 insects-15-00590-f004:**
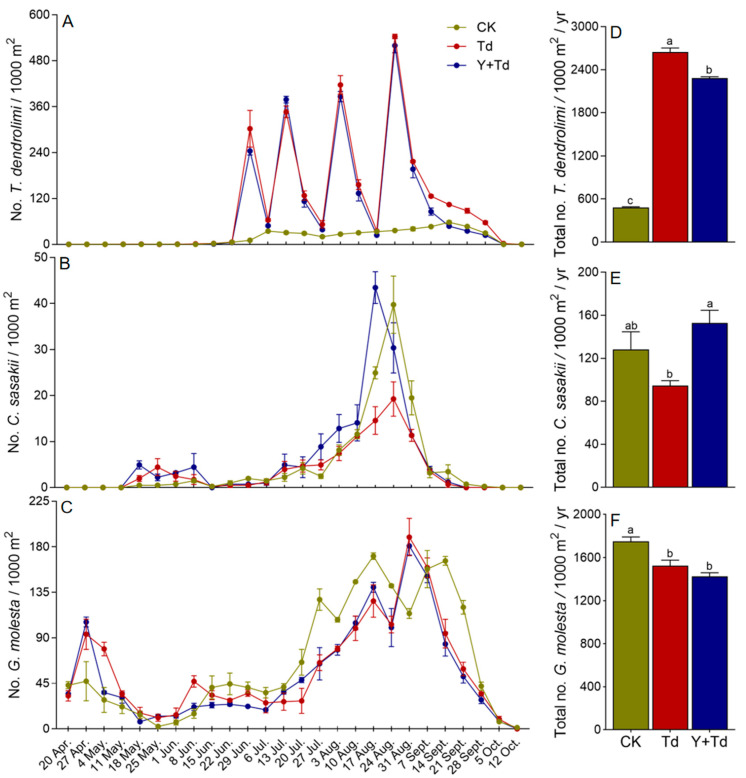
Weekly trap catches of *T. dendrolimi* (**A**), *C. sasakii* (**B**), and *G. molesta* (**C**) and cumulative trap catches (**D**–**F**) in different treatment blocks in a pear orchard. CK = control, Td = *T. dendrolimi* releases, and Y + Td = yellow sticky traps plus *T. dendrolimi* releases. Means bearing different letters were significantly different among treatments (one-way ANOVA followed by Tukey’s test, α = 0.05).

**Figure 5 insects-15-00590-f005:**
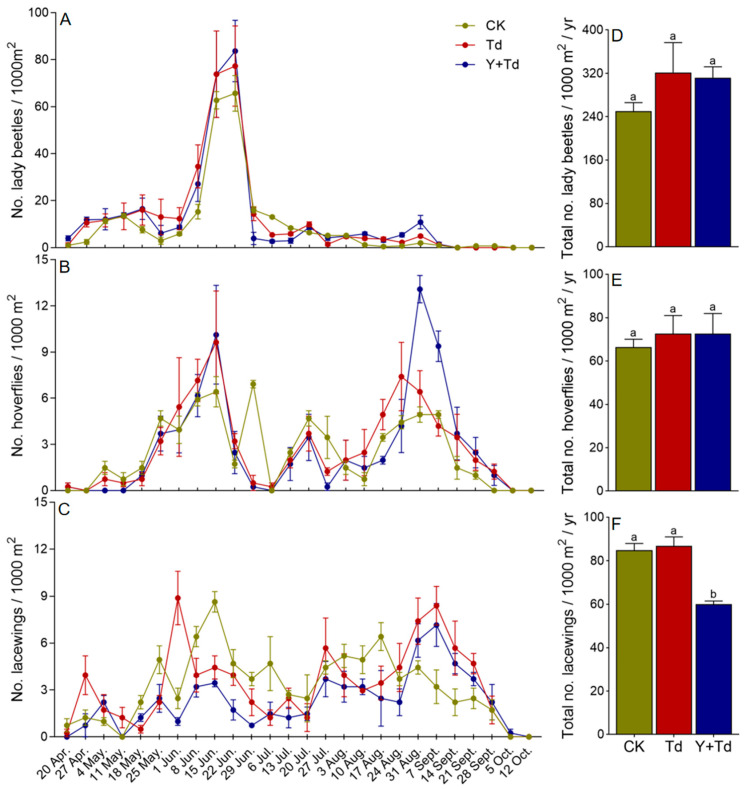
Weekly trap catches (**A**–**C**) and cumulative trap catches (**D**–**F**) of lady beetles, hoverflies and lacewings in different treatments (CK = control, Td = *T. dendrolimi* releases, Y + Td = yellow sticky cards + plus *T. dendrolimi* releases). Means bearing different letters were significantly different among treatments (one-way ANOVA followed by Tukey’s test, α = 0.05).

## Data Availability

The data presented in this study are available on request from the corresponding author (ZL).

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
