# Peer review of "Yellow Sticky Cards Reduce the Numbers of Trichogramma dendrolimi (Hymenoptera: Trichogrammatidae) Following Augmentative Releases against the Fruit Borers Carposina sasakii (Lepidoptera: Carposinidae) and Grapholita molesta (Lepidoptera: Tortricidae) in a Pear Orchard"

_insects, 2024, doi:10.3390/insects15080590_

Round 1
Reviewer 1 Report
Comments and Suggestions for Authors
The manuscript "Yellow sticky cards reduce the numbers of Trichogramma dendrolimi..." is very well written, detailing a set of experiments which are robust and applicable to pest management in pome fruit. Although it is not common practice where I am from to use yellow sticky cards for management at an orchard scale, I appreciated the research and the recommendation to combine strategies to best advantage against the pest while conserving the natural predators. I found very few issues which require attention and have detailed them below. I recommend accept with minor revisions.
Summary/Abstract
L24 – ‘relies upon the mutual…’
L26-27 – ‘conducted color choice tests’ … ‘with yellow sticky cards’… unclear how it is a color choice test if yellow is being singled out? Were there 2 sets of lab/field trials?
L33 – suggest reversing this sentence ‘The cards did not capture high numbers of lady beetles, but did capture adult lacewings, likely reducing their numbers in the field.’
Introduction, pg. 2
L2 – ‘effective approach for sustainable…’ or ‘effective approach to achieve sustainable…’
L3 – ‘promotes the use of combinations of….’
L4 – ‘tactics which can include plant…’
L13 – I think there should be commas between 12, 13, 14 ?
L29, 31, 43 and 48, 51, 54, – see comment for L13
L43 – I don’t think you need ‘within plants’ as you’ve already indicated that they ‘bore into fruits’ earlier in the sentence
Introduction, pg. 3
L4 – is this for control or just monitoring? Control would mean much longer time in the orchard than monitoring
L6-14 – the abstract speaks of ‘color choice tests’ but the objectives are focussed on the yellow sticky cards only?
L26 – where were the boxes sourced from?
L29 – the plastic boxes were destroyed during this process?
L40 – why this ratio of male to female?
Methods, pg. 4
L12 – light intensity? This wasn’t mentioned in the objectives
L14 – do you mean ‘One assay compared 5 light intensities and a second assay compared 4 colors’?
L38 – ‘trees’
Methods, pg. 5
Figure 1 – need to add abbreviations to the legend, e.g. ‘T. dendrolimi (Td)’ and clarify the ‘Ck’ – control? No T. dendrolimi?
L5 – how far away from the release point were the yellow sticky cards placed? It would seem about 7 m?
L10 – an extra comma after ‘pheromone lure’ can be deleted
L11 – how did you determine ‘peak’ if traps were checked weekly? Would the T. dendrolimi be released the week captures started to decline?
L21 – how large were the sticky cards?
Methods, pg. 6
L2 –So there are 2 experiments here: one with 6 colors, and then a second experiment with the two most preferred colors (from the 6-color experiment)? Not entirely clear when read the first time that this is what happened. Just need a connector sentence like ‘… sticky card was tallied. Based upon the results from the 6-color experiment, we then conducted a two-way choice tests for each species….’
L3 – were there 2 sticky cards at each end of the cage (1 card of each color and 4 cards in the cage)? Or a single card (of 1 color) at each end (thus 2 cards in the cage)?
L4 – suggest clarifying more specifically whether the data is coming from the lab or field. For the chi-square was the assumption of equal distribution across light intensities and colors? Should state this.
L5 – which tests were done in Excel? And which in SPSS?
L11 – ‘…compared using one-way ANOVA followed by Tukey’s HSD mean separation test.’
Results, pg. 6
L10 – did you use any transformation (e.g. Log (x + 1)) on the data to ensure normality? Did you test to ensure your data met the assumptions for ANOVA analysis?
L14 – section title needs to capture the color choice test as well
Figure 2 – ‘Choice tests…’ as there were more than 1.
Results, pg. 7
Figure 3 – ‘Choice tests…’
L9 – ‘sentinel sticky cards’? there is no mention of these in the Methods on pg. 5. Do you mean the ‘yellow sticky cards for monitoring’? what’s the difference between the cards themselves – just the number deployed?
Results, pg. 8
Figure 4 – are the weekly trap catches from the ‘sentinel sticky cards’ only? Need to clarify this.
L8 – are these likewise tallied only from the sentinel sticky cards?
L12 – ‘…no significant difference…’
Discussion, pg. 9
L7 – end sentence after ‘foraging’ and start new thought ‘Colored sticky cards have…’
Discussion, pg 10
L6 – commas needed in reference numbers
L25 – not sure ‘compensated’ is the correct term here. I think ‘substituted’ or ‘replaced’ might be better. What’s happening is that both T. dendrolimi and G. molesta are being reduced by the yellow sticky traps – either of these methods would serve to provide some relief from damage by this species. On L45 you use ‘mitigated’ and this is likely the better word choice in this case.
Reviewer 2 Report
Comments and Suggestions for Authors
The paper is generally well written. A few places where there could be minor English editing for clarity. Figures are clear and easily interpreted. The F statistic usually has 2 accompanying values for degrees of freedom. The F statistics in this paper only have a single value for degrees of freedom. Throughout the text where reference numbers were listed in brackets, there were no commas between the individual reference numbers. Example: [231332] instead of [23, 13, 32].

Comments on the Quality of English LanguageThe English is very good. I found several places where changes could be made for clarity.
Reviewer 3 Report
Comments and Suggestions for Authors
The paper reports an interesting experiment on the influence of commonly used Integrated Pest Management (IPM) strategies, specifically the use of color sticky traps, on the presence and release of parasitoids. While the paper is fairly well-written, it requires some adjustments in the introduction and additional detail in the materials and methods section. The introduction, in particular, appears fragmented and would benefit from being revised into a cohesive and smooth narrative.
P2, Line 13, revise reference
P2, Line 26, this part is not well linked with the one above, the colored sticky traps should be better introduced and one of the component of IPM for mass trapping and/or monitoring pest species.
P2, Line 39, same. This part is not well linked with the one above.
P3 line 3, revise this sentence, because the yellow cards were already introduced before.
P4, I don’t really understand the part regarding the light intensity choice if the paper is related to the different sticky trap and their capture.
In the same section, how many times the experiments were replicated?
P4, line 47, once pheromone traps were used, here the authors should give more details: who was the target, the type of trap, the pheromone molecule, the doses if known, etc
P6 the order of the materials and methods should follow the same order of the results and viceversa.
P9, line 11,12 …but they also orient to white.
P10, line 15,16 also mention the costs problem…
Comments on the Quality of English Languageno comment
